# MISTILL: DISTILLING DISTRIBUTED NETWORK PROTOCOLS FROM EXAMPLES

## ABSTRACT

New applications and use-cases in data center networks require the design of Traffic Engineering (TE) algorithms that account for application-specific traffic patterns. TE makes forwarding decisions from the global state of the network. Thus, new TE algorithms require the design and implementation of effective information exchange and efficient algorithms to compute forwarding decisions. This is a challenging and labor and time-intensive process. To automate and simplify this process, we propose `Mistill`. `Mistill` distills the forwarding behavior of TE policies from exemplary forwarding decisions into a Neural Network. `Mistill` learns how to process local state to send it over the network, which network devices must exchange state with each other, and how to map the exchanged state into forwarding decisions. We show the abilities of `Mistill` by learning three exemplary policies and verify their performance in simulations on synthetic and real world traffic patterns. We show that the learned protocols closely implement the desired policies, and generalize to unseen traffic patterns.

## 1 INTRODUCTION

Data center workloads are diverse and range from client-server applications, machine learning, and web-traffic to high-performance computing applications (Benson et al. (2010); Roy et al. (2015); Zhang et al. (2017)). Each workload has different requirements towards Traffic Engineering (TE). For example, optimizing the completion time of individual flows is desirable for client-server applications (Dukkipati & McKeown (2006)), but leads to decreased performance in machine learning workloads (Chowdhury et al. (2014)). To achieve the best performance, each workload needs a TE scheme that is tailored towards its specific requirements. Indeed, new applications regularly trigger the development of new TE schemes in the networking community (Katta et al. (2016); Chowdhury et al. (2014); Chen et al. (2018); Benson et al. (2011)).

Obtaining a deployable distributed protocol from an initial TE design is challenging. A design includes the specification of update messages, the processing of update messages, and the algorithm to compute the forwarding decisions that result in the desired policy (Katta et al. (2016)). The design of each component is challenging since the final protocol must meet line-rate, exchange information at (sub)millisecond granularity, and cope with limited computational requirements. All steps must be efficient, and the design holistic, requiring the exploitation of patterns in the network and the traffic. For instance, `CONGA` integrates a substantial part of the calculation for the forwarding decision into the exchange of update messages (Hsu et al. (2020)). The complexity makes customizing TE schemes hardly feasible for smaller enterprises, potentially resulting in competitive disadvantages.

To automate and simplify the translation of a TE policy to a distributed protocol, we propose `Mistill`. `Mistill` distills the forwarding behavior of a TE policy from exemplary forwarding decisions into a Neural Network (NN). `Mistill` learns the encoding of the switch local state in update messages, the exchange of update messages, and the computation of forwarding decisions from update messages with Machine Learning (ML). `Mistill` removes the need to manually design update messages, information exchange, and the calculation of forwarding decisions. ML allows `Mistill` to automatically detect helpful patterns in traffic and network and exploit the patterns to learn a protocol. Further, our NN design makes the learned patterns accessible to humans and can thus aid the manual design process.

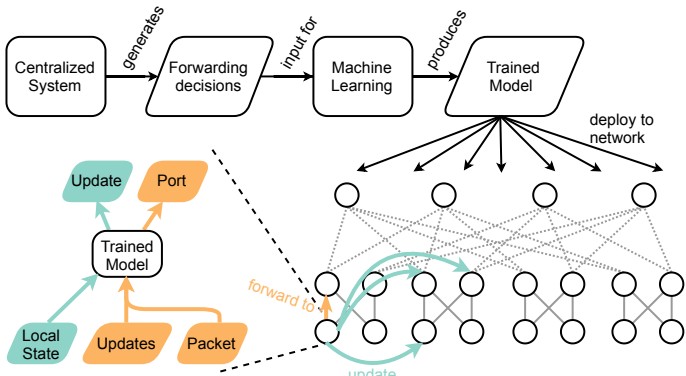

Figure 1: `Mistill` trains a NN with forwarding decisions from an initial centralized TE system. `Mistill` deploys the trained NN to the switches, which query the NN for forwarding decisions.

Our main contributions in this work are: 1) The design of an ML model that can learn forwarding behavior, makes learned patterns accessible to humans, and has the potential to be deployed on real hardware. 2) We show in simulations that `Mistill` can learn the forwarding behavior of three TE policies and generalizes to previously unseen traffic patterns. 3) We analyze the learned messages and their exchange. We find that `Mistill` can learn distributed protocols that closely implement the desired policy and are robust towards distributional shift in the input. We analyze the learned messages' content, and show that the NN learns an information exchange resembling an edge-cover, which can be used to optimize the information exchange of other TE schemes.

The document is organized as follows. Sec. 2 describes the application scenarios and the requirements towards ML models. Sec. 3 describes the neural architecture and justifies design choices. Sec. 4 describes experiments and presents results and visualizations of the learned protocols. Sec. 5 discusses related work, and Sec. 6 concludes the paper.

## 2 APPLICATION SCENARIO

`Mistill` learns a distributed protocol from examples. Fig. 1 illustrates the process and expected outcome. The first component is a centralized implementation of a TE policy. For example, a packet or flow level simulation using an Integer Linear Program or heuristic algorithm to compute forwarding decisions based on the entire network state. The forwarding decisions specify how a switch forwards traffic to its neighbors, given the current state of the network. Usually, new TE schemes are evaluated in such a setting first to establish their performance and correctness.

Here, `Mistill` hooks in and uses the forwarding decisions to train an ML model, i.e., a NN. `Mistill` trains the model offline, i.e., *not* on the network devices. The NN learns three aspects: 1) how to encode local state in update messages, 2) how to exchange the update messages, 3) how to map the exchanged state into forwarding decisions.

After training, `Mistill` deploys the NN to switches, e.g., with P4 (Siracusano & Bifulco (2018)). All switches in the network have *the same* NN and use it in inference mode, i.e., do forward passes only. The NN distinguishes switches through an identifier. During training, the NN learns how to condition the forwarding decisions on the identifiers, thus learning the correct behavior for each switch. To compute forwarding decisions, switches do not need to know the network topology.

This work does not try to outperform existing TE strategies in their specific application scenarios. Instead, this work aims to facilitate the automatic generation of novel TE policies, taking direct advantage of switch architectures like Taurus (Swamy et al. (2020)) without asking network operators to take the time or acquire the expertise to design highly optimized protocols and implement them, e.g., in P4 (Bosshart et al. (2014)).

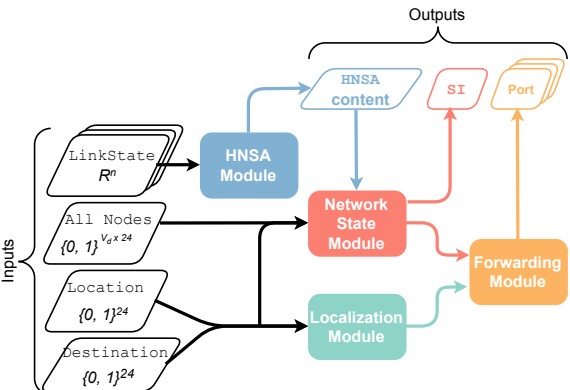

Figure 2: The NN architecture consists of four modules: The LocMod, the HNSAMod, the NSMod and the Fwd Mod.

## 2.1 Data center network requirements

The goal of `Mistill` is to distill network protocols for data-center networks from exemplary data. We focus on Clos topologies due to their popularity and simple structure (Zhang et al. (2019)). See Appendix B for details. TE protocols in data-centers exchange update messages at (sub-)millisecond scale (Katta et al. (2016)). Thus, the updates must be small to keep the resulting overhead low. To effectively utilize the frequent updates, switches must make forwarding decisions frequently, e.g., for each new flow. Thus, the decision logic should reside in the data-plane (Alizadeh et al. (2014)). This limits the computational demand the decision logic can express towards the switch. The learned decision logic must thus be computationally efficient to meet the limited computational capacities.

## 2.2 Why Neural Networks?

`Mistill` uses a NN to learn a distributed protocol, which is a good choice for three reasons.

1) NNs are general function approximators (Glorot et al. (2011)). This is important because forwarding decisions can depend on the network state in a non-linear way. For example, minimizing the maximum link utilization results in forwarding decisions that do not depend linearly on the network state. A small change in the utilization can lead to a sudden change in the forwarding decision.

2) NNs can be executed with the limited computational resources in the data-plane (Sanvito et al. (2018); Swamy et al. (2020)). Here, the choice of activation function is important. The $\mathrm{ReLU}$ activation requires only a check for negativity (Glorot et al. (2011)). A NN with $\mathrm{ReLU}$ activations uses addition, multiplication and a check for negative values. Further, NNs can be quantized to integer or binary computations (Gholami et al. (2021)).

3) The execution of NNs is easy to parallelize and accelerators for NN inference already exist for network equipment (Luinaud et al. (2020); Swamy et al. (2020)). In addition, research efforts show that NNs can be executed in the data-plane at line rate already on today's ASICs (Siracusano & Bifulco (2018)), and on end-hosts with SmartNICs (Sanvito et al. (2018)). We thus believe that switches will be increasingly capable of executing NNs at line rate.

## 3 Neural Architecture

The NN architecture consists of four modules: the Localization Module (LocMod), the HNSA Module (HNSAMod), the Network State Module (NSMod), and the Forwarding Module (Fwd Mod). Fig. 2 shows how the modules interact and represents inputs and outputs with hashes. See Appendix C for sketches of technical realizations of `Mistill`.

**Output.** The NN has three outputs: the out-port, Hidden Node State Advertisements (HNSAs)[1], i.e., update messages that inform other network devices about the local state of the sending device, and State Interest (SI), signaling which devices need to exchange information with each other.

**Input.** The NN has four inputs: `LinkState`, `AllNodes`, `Destination` and `Location`. `LinkState` is the switch local state, e.g., the availability and utilization of incident links. The NN transforms the `LinkState` into HNSAs. `Location` is the identifier of a switch, `Destination` the destination of a flow, and `AllNodes` the identifiers of all switches. The NN uses `Location`, `Destination`, and `AllNodes` to learn how to condition forwarding decisions on a specific switch and destination and which switches' HNSAs are needed to make a forwarding decision.

**Training.** The NN is trained end-to-end. Training end-to-end tailors HNSAs and SI to a forwarding policy. The loss function depends on the forwarding objective. We interpret objectives as multi-label classification or as learning of a multinomial distribution. We use: $\mathcal{L}_{mn} = -\sum_{i=1}^{N} p(x_i) \log p(y_i)$ as loss function for the multinomial distribution, and $\mathcal{L}_{ml} = -\sum_{i=1}^{N} (p(x_i) \log p(y_i) + (1 - p(x_i)) \log(1 - p(y_i)))$ as loss function for the multi-label classification. $N$ corresponds to the number of outputs, $p(x_i)$ is the ground truth and $p(y_i)$ is the prediction of the NN. The loss $\mathcal{L}_{mn}$ can learn distributional targets to split traffic over adjacent nodes. $\mathcal{L}_{ml}$ can learn policies that have one or multiple equally good paths. Changes in the network that lead to changes in the forwarding policy require a re-training of the model. For example, new networking hardware, or changes to the physical network topology. As we will show, re-training is short compared to the deployment process of new hardware, and can be integrated into the roll-out process.

**Inference.** After training finishes, the NN is deployed to the switches. The switches use the NN for three tasks: 1) to compute HNSAs from their local state, 2) to select the HNSAs necessary to calculate the out-port, and 3) to calculate the out-port. The switches do not change the weights of the NN. The switch also never execute the full NN. To compute HNSAs, the switch executes the HNSAMOD. To compute the out-port, the switch executes the LocMOD, NSMOD and FWD MOD, but not the HNSAMOD. The switch uses the HNSAs it received from other nodes instead.

### 3.1 DATA FORMAT

The inputs `Location`, `Destination` and `AllNodes` are binary vectors in $\{0, 1\}^{24}$ and are the last three octets of IPv4 addresses following the addressing scheme for Fat-Trees (Al-Fares et al. (2008)). We experimented with different node representations and enclose the results in Appendix D.

The `LinkState` $\in \mathbb{R}^{6}_{\geq 0}$ is a real-valued vector and represents incident edges. The first four attributes are a one-hot encoding of the link availability in each direction. For example, `10 10` means that both directions are up, and `10 01` means that one direction is up and one is down. The fifth and sixth attributes are the weight of the link in each direction, representing, e.g., link utilization. The link state of a switch is the concatenation of the vectors of all its links, resulting in a vector from $\mathbb{R}^{6 \cdot \hat{d}}_{\geq 0}$, where $\hat{d}$ is the maximum degree of switches in the network. Vectors of switches with a smaller degree are zero-padded to this length.

### 3.2 LEARNING THE HNSA CONTENT

The HNSAMOD transforms the `LinkState` of a switch into a binary vector. The binary vector is the content of a HNSA . The vector corresponds to a hidden layer with binary activations. Since switches send the HNSA over the network, binary activations are better than real-valued activations, since a binary representation allows the NN to have more neurons for the same number of bits. For example, a binary vector of 128 bits corresponds to four floating-point numbers. Since NNs store information in a distributed representation, having more neurons with low precision is better than having fewer neurons with high precision (Glorot et al. (2011)).

We sample the binary activations from $m$ independent, one-hot encoded categorical distributions of arity $n$: HNSA $\sim \mathrm{Cat}(n, \theta_1) \, || \, \ldots \, || \, \mathrm{Cat}(n, \theta_m)$. The NN calculates the parameters $\theta_i$ from the `LinkState` by passing the `LinkState` through a feed forward NN with one hidden layer and

---

[1]In analogy to Link State Advertisements (LSAs) in link-state routing protocols. *Hidden* refers to the fact that the content of the messages is the activations of a hidden layer of the NN. Further, HNSAsdiffer from LSAs in that they are not per link but per node.

ReLU activation. To keep the NN differentiable, we use a re-parameterization trick to sample from the categorical distributions (Jang et al. (2017); Maddison et al. (2017)). During training, the NN learns how to parameterize the categorical distributions from the `LinkState` to encode information in the binary hidden layer. The arity $n$ and number of distributions $m$ are hyperparameters of the NN architecture and optimized during training.

## 3.3 LEARNING TO COMMUNICATE

The NSMOD learns: 1) the SI output, i.e., whose `HNSAs` are needed to make a forwarding decision on a specific switch and destination, and 2) how to process `HNSAs` to make forwarding decisions.

To learn the SI and process `HNSAs`, we use an attention mechanism inspired by the scaled dot-product attention (Vaswani et al. (2017)), which can be interpreted as a learned lookup table that retrieves `HNSAs` based on the switch that executes the NN and a given destination. The attention mechanism has three inputs: Keys $K$, Queries $Q$ and Values $V$. $K$, $Q$ and $V$ are matrices, where $K$ and $V$ must have the same number of rows (Vaswani et al. (2017)). $K$ and $Q$ are used to compute attention-weights $A$, with which a convex combination of $V$ is computed. The attention scores $A$ are $A = \text{gumbelSoftmax}\left(\frac{1}{\sqrt{r}} W_q Q \cdot (W_k K)^T\right)$, where $W_q$ and $W_k$ are a trainable linear transformation of $Q$ and $K$, $r$ is the number of rows of $K$, and gumbelSoftmax is the gumbel softmax reparameterization trick (Jang et al. (2017); Maddison et al. (2017)). The output of the attention layer is $A \cdot (W_v V)$, where $W_v$ is a trainable linear transformation. $Q$ corresponds to the concatenation of `Destination` and `Location`, $K$ to `AllNodes` and $V$ to `HNSAs`. We use Multi-Head Attention (MHA), i.e., parallel attention mechanisms, and concatenate their output. The concatenated output is passed through a hidden layer with ReLU activation, and then into the FWD MOD.

The SI are the attention scores of the individual attention heads. Non-zero entries signal that the corresponding `HNSA` contributes information for the forwarding decision. The scores $A$ make learned patterns explicit and interpretable. To enforce sparse patterns, we use the gumbelSoftmax trick to calculate attention scores (Jang et al. (2017); Maddison et al. (2017)). Thus, each attention head selects one `HNSA` . The number of attention heads is an upper bound on the exchanged messages.

## 3.4 LEARNING TO FORWARD

The forwarding decision of the NN is based on the network state encoded in `HNSAs`, i.e., the output of the NSMOD, the destination and the switch that makes the forwarding decision. The NN learns to combine this information to react to changes in the state of the network, and to learn how forwarding decisions differ for switches and destinations.

The calculation of the out-port requires the execution of the NSMOD, the LOCMOD and the FWD MOD. The LOCMOD and the FWD MOD each consist of one feed forward NN with one hidden layer and ReLU activation.

## 4 EXPERIMENTS

`Mistill` learns a TE policy protecting the data-center network from traffic whose destination is not reachable. If no route to a host exists, then traffic is dropped. The challenge lies in accounting for link failures in the up-link from hosts to switches, which existing TE schemes cannot handle. Dropping traffic can be important for applications that send larger amounts of data with the User Datagram Protocol (UDP). Existing TE schemes would forward this traffic to the last hop of the network and thus waste bandwidth (Hsu et al. (2020); Dutt (2017)). If a route exists, we consider three policies to forward traffic: 1) `WCMP`: Splits traffic over neighbors in proportion to the available down-stream bandwidth, 2) `LCP`: forwards traffic along the path with the least weight, and 3) `MinMax`: Forwards traffic along the path with the smallest maximum link utilization. All policies should work in the presence of arbitrary node and link failures. We compare the quality of the learned protocols to Equal Cost Multi-Pathing (ECMP) and the actual forwarding decision.

| | HNSAMOD | | | LOCMOD | NSMOD | | | | | FWD MOD |
| | FCN | #Distr. | Arity | FCN | #Heads | $W_q$ | $W_k$ | $W_v$ | FCN | FCNs |
|---|---|---|---|---|---|---|---|---|---|---|
| MinMax | 108 | 64 | 2 | 90 | 14 | 48 | 41 | 17 | 109 | $104, 84$ |
| LCP | 63 | 64 | 2 | 90 | 14 | 48 | 48 | 50 | 125 | $102, 105$ |
| WCMP | 48 | 64 | 2 | 90 | 14 | 62 | 62 | 27 | 110 | $76, 71$ |

Table 1: Hyper parameters for the best models.

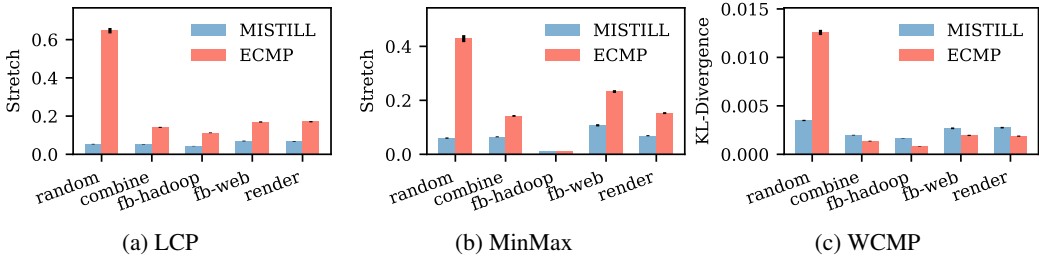

(a) LCP      (b) MinMax      (c) WCMP

Figure 3: Bar plots for TE policies and traffic patterns.

## 4.1 DATA GENERATION AND NN TRAINING

We generate data for a $k = 8$ Fat Tree having 128 hosts and 80 switches (Al-Fares et al. (2008)). To make the protocols robust to node and link failures, we remove between zero and ten edges and zero and five nodes uniformly at random from the topology. For the remaining edges, we sample edge weights from a uniform distribution. We then sample a source and destination pair and compute each policy's forwarding decisions on the generated graph for the sampled pair.

The training set consists of $200\,000$ problem instances, the validation set of $50\,000$ problem instances. The final results are obtained from a packet-level simulation. We use the Asynchronous Successive Halving Algorithm (ASHA) (Li et al. (2018)) implemented in the Ray Tune (Liaw et al. (2018)) library for hyperparameter optimization and select the best configuration for the evaluation. Tbl. 1 lists the hyperparameter of the best performing models. For all policies, the HNSAMOD consists of one fully connected layer with ReLU activation and a linear transformation that generates the logits for the gumbelSoftmax function. The LOCMOD consists of one fully connected layer with ReLu activation. The NSMOD consists of one MHA module and the FWD MOD of two fully connected layers. We describe the training process and the hyperparameter search space in detail in Appendix A. Training a single model takes in the order of 5 hours on an NVIDIA Tesla V100 GPU. Training 100 models in parallel with ASHA on three NVIDIA Tesla V100 GPUs takes 60 hours. We implemented the NN in C and CUDA to evaluate the inference time and evaluate the LCP NN, the largest network, on an Intel(R) Core(TM) i7-7700HQ CPU with $2.80\,\mathrm{GHz}$, and an NVIDIA Tesla V100 GPU. Inference time on a single CPU core without parallelization is $4\,\mathrm{ms}$ and $0.047\,\mathrm{ms}$ on the GPU, which suffices to meet the inter-arrival time of flows in data-centers (Benson et al. (2010)).

## 4.2 ROUTING PERFORMANCE

We evaluate five traffic patterns and perform 100 simulations with random node and edge failures for each pattern. Each simulation samples between zero and ten edge failures and between zero and five node failures. The traffic patterns are random, combine and render (Delimitrou et al. (2012)), and fb-hadoop and fb-web (Roy et al. (2015)). In random, the utilization of each edge is sampled from a uniform distribution. The other patterns are created from traffic matrices of data-center applications. fb-hadoop is based on a Hadoop cluster in a Facebook data-center, fb-web is based on the production traffic of a Facebook data-center (Roy et al. (2015), combine results from a web-search, and render renders the search results for users (Delimitrou et al. (2012)). Note that combine, render, fb-hadoop and fb-web result in utilization patterns in the network that strongly differ from the random utilization values the NNs were trained with. In a simulation, packets traverse the network from each ToR to every server. For each packet, we compare the metric of the optimal path with the metric of the path from the NN and Equal Cost Multi Pathing (ECMP).

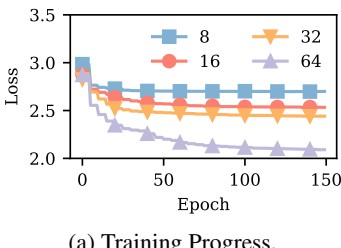
(a) Training Progress.

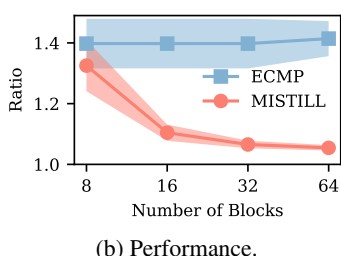
(b) Performance.

Figure 4: Training loss for varying number of blocks in the GS for the calculation of `HNSA`s.

**LCP:** Fig. 3a shows bar plots for the `LCP` policy, and compares `Mistill` to `ECMP`. The Y-Axis shows the average stretch of the objective, i.e., the ratio that `Mistill` and `ECMP` are worse than the optimal solution. Error bars correspond to the 99 % confidence interval of the mean. Fig. 3a shows that `Mistill` is better for all traffic patterns, with paths whose weight is close to the optimal weight. Further, Fig. 3a shows that the error of `Mistill` is similar across all traffic patterns. The NN thus generalizes to distributions of the inputs that it has never seen during training.

**MinMax:** Fig. 3b shows a similar plot as in Fig. 3a for the `MinMax` policy. Fig. 3b shows that `Mistill` is always better than `ECMP`, except for `fb-hadoop`. For `fb-hadoop`, `Mistill` and `ECMP` are both almost zero. This is due to the specific pattern in `fb-hadoop`, in which the up-links of hosts are the most loaded links, and the TE policy has thus almost no impact on the objective. As for `LCP`, the NN generalizes to unseen input distributions.

**WCMP:** We use the Kullback-Leibler Divergence (KLD) to evaluate `WCMP`. Fig. 3c shows the average KLD for `Mistill` and `ECMP` for the traffic patterns. Error bars correspond to the 99 % confidence interval of the mean. Fig. 3c shows that `Mistill` is better than `ECMP` for the `random` traffic pattern. `ECMP` has slightly smaller average KLDs for the other patterns. This is because available bandwidth is almost equally distributed over paths. `WCMP` essentially becomes `ECMP` up to a small error. Note how the average KLDs of `Mistill` decreases for the other traffic patterns. This indicates that the NN also generalizes in this case to unseen input distributions.

In summary, our experiments show that `Mistill` can learn the forwarding behavior of three popular TE policies from exemplary samples. Further, the results show that the NN generalizes to previously unseen input distributions.

### 4.3 HNSA SIZE

We investigate the impact of the number of categorical distributions on the `MinMax` policy and vary them in $\{8, 16, 32, 64\}$. Fig. 4a shows that more distributions results in a smaller loss, and that the decrease in loss is strong during the initial stages of training and converges towards the end.

Fig. 4b compares the ratio between `ECMP` and the true solution to the ratio of `Mistill`. The solid line corresponds to the mean, the shaded area to the mean's 99 % confidence interval. Fig. 4b shows that 8 distributions result in a protocol that performs better than `ECMP`. The protocol with 8 distributions, i.e., 8 bits, mostly accounts for link failures and little for edge weights. The performance of `Mistill` improves with more distributions and approaches the ground truth and the confidence interval shrinks. The improvement between 8 and 16 distributions is larger than between 16 and 64. With 64 distributions, `Mistill` selects paths with similar metrics to the ground truth. With 64 distributions, the probing overhead in a network supporting up to 1 024 hosts with no over subscription and millisecond updates is 1.22 % for 10 Gbits links, i.e., in a similar order as existing TE schemes (Katta et al. (2016)). See Appendix C for details.

### 4.4 VISUALIZATIONS

This section visualizes the learned `HNSA` messages and the attention weights. We compare the attention weights of the gumbelSoftmax activation with the weights of a model trained with the sparsemax activation function (Martins & Astudillo (2016)) for the `MinMax` policy.

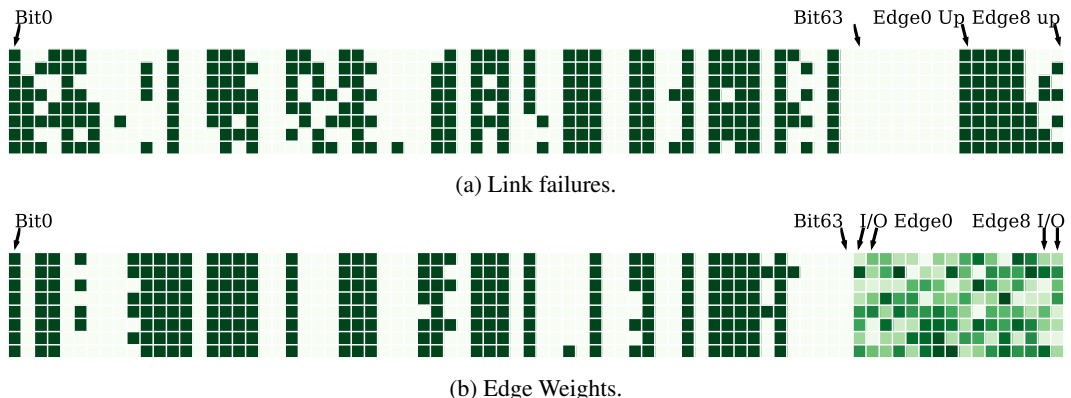

(a) Link failures.

(b) Edge Weights.

Figure 5: Bit pattern in `HNSA` messages for edge availability and edge weights.

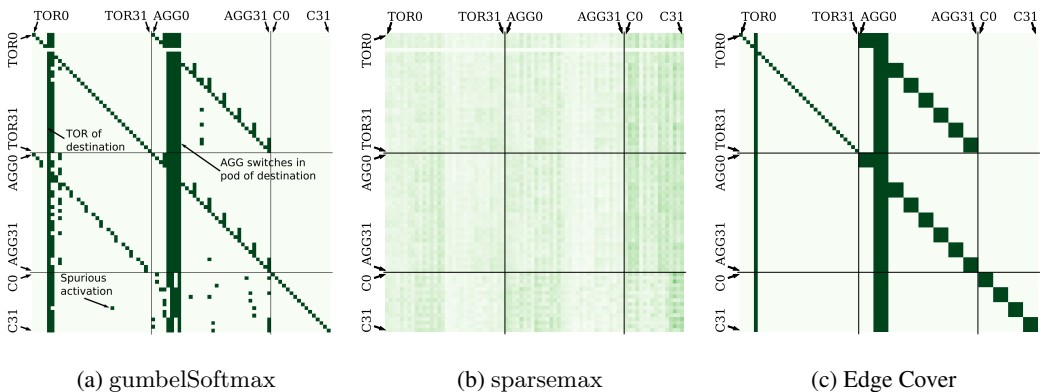

(a) gumbelSoftmax

(b) sparsemax

(c) Edge Cover

Figure 6: Attention weights. Each row contains the weights for one switch over all other switches.

**HNSA content**   Fig.5 shows `HNSAs` of `Mistill` for the `MinMax` policy. Fig. 5a illustrates how link failures reflect in the `HNSAs`. Fig. 5b illustrates how edge weights reflect in the `HNSAs`. The left part of Fig. 5a and Fig. 5b show the `HNSAs`. The right part shows the availability of edges in Fig. 5a, and the edge weights in Fig. 5b. White corresponds to a value of zero, green to a value of one.

Fig. 5a shows that a large portion of the bits encode the link availability. In fact, 56 out of 64 bits can be associated with link availability. Some bits even directly correspond to the availability of edges. `Bit59` and `Bit26` corresponds to `Edge8`, and `Bit4` to `Edge7`.

Consequently, Fig. 5b shows that most bits are constant for different edge weights. The NN uses eight out of the 64 bits to encode edge weights. The relation between bits and edge weights is more complex than between link availability and bits, and no clear correspondence is observable.

**Attention Scores**   Fig. 6 illustrates the attention scores of all switches in the network for a specific destination node. To evaluate the effect of the gumbelSoftmax activation, we additionally trained a NN with the sparsemax activation function (Martins & Astudillo (2016)). Further, we handcrafted attention scores corresponding to an edge cover of all edges belonging to the shortest paths between a switch and the destination. Fig. 6a shows the attention scores of the gumbelSoftmax activation, Fig. 6b of the sparsemax activation, and Fig. 6c the edge cover. The attention scores of individual attention heads are summed up and clipped to a maximum value of one.

Fig. 6a shows that the attention scores of the gumbelSoftmax resemble the edge cover in Fig. 6c. In contrast, the attention scores of the sparsemax cover all nodes. Fig. 6a show that the NN relies on the state of the ToR switch- and the aggregation switches in the pod of the destination. Further, all switches rely on their state. Some activations in Fig. 6a are not intuitive. For example, it is unclear why core switch C10 relies on the state of a ToR switch in a different pod than the destination.

Fig. 6a and 6b show that the `sparsemax` activation results in a dense pattern that results in a large signaling overhead of the learned protocol. The resulting protocol would exchange one message for each non-zero entry. The `gumbelSoftmax` activation has a clear advantage and requires exchange of at most 10 messages, which is in the order of the edge cover, which requires at most 9 messages.

In summary, the majority of the bits in the `HNSAs` encode the edge availability. The attention scores resemble an edge cover of edges that belong to the shortest paths to a destination. The detected pattern is sparse and can be used to optimize the sending of update messages. This behavior does also hold for the `WCMP` and `LCP` policy.

## 5 RELATED WORK

**Routing and Traffic Engineering.** TE solution for DC networks addressing different goals and challenges exist (Noormohammadpour & Raghavendra (2017)). Solutions range from using well established protocols such as OSPF Chiesa et al. (2016); Michael & Tang (2014) over new centralized designs (Al-Fares et al. (2010); Benson et al. (2011); Zhang et al. (2016)) to distributed protocols that operate directly in the data plane (Hsu et al. (2020); Kandula et al. (2007); Alizadeh et al. (2014); Katta et al. (2016)). Closest to `Mistill` is CONTRA (Hsu et al. (2020)), which proposes a high-level policy language expressing TE goals, and uses a path-vector based protocol for network state exchange. In contrast to previous work, `Mistill` is *not* yet another TE scheme and does not aim to outperform existing TE schemes in scenarios they are optimized for. Instead, `Mistill` is a method to automate the design of distributed protocols by distilling protocols from data. In contrast to CONTRA, `Mistill` does not require a policy language. Further, `Mistill` is not limited by the capabilities of a path vector protocol. Instead, `Mistill` learns the content, the exchange and the processing of information from data, and automatically detects useful patterns.

**ML and Networking.** Combinations of ML and networking have been explored from different angles in the past (Boutaba et al. (2018)). Closest to `Mistill` are work from (Geyer & Carle (2018)) and (Xiao et al. (2020)), which rely on Recurrent Neural Networks to learn forwarding behavior. In contrast to previous work, `Mistill` focuses on data-center applications. Proposals in (Geyer & Carle (2018); Xiao et al. (2020)) focus on wide area networks and do not apply to data centers. Both proposals require the exchange of several messages between neighboring nodes, similar to a distance-vector algorithm, to accommodate a change in the network. This procedure is too slow for the fast-changing traffic conditions in data-center networks. `Mistill` does not suffer from this problem and maps state updates directly into forwarding decisions. Further, we are the first to design such a system with practical deployment in mind. Key parts of the design of `Mistill` can, in principle, be deployed in the network, e.g. using Taurus (Swamy et al. (2020)).

## 6 CONCLUSION

This paper investigates if it is possible to learn a distributed protocol in a data-driven manner that implements a custom TE policy. Accordingly, this paper proposes `Mistill`, an ML-based approach that learns distributed protocols from exemplary forwarding decisions. `Mistill` learns the processing and exchange of local information with other network elements and the computation of forwarding decisions from the exchanged information. Further, we design `Mistill` with deployment on hardware in mind. `Mistill` uses only calculations that are available on network devices.

To show the applicability of `Mistill`, we learn the forwarding behavior of three TE policies and evaluate the learned protocols in simulations on traffic patterns from four realistic data-center applications. Our results show that it is possible to learn distributed protocols from data that closely match the original forwarding decisions and generalize to previously unseen traffic patterns. Further, we analyze the learned representations and show that they are reasonable. In the appendices, we provide detailed information on how `Mistill` can be implemented in practice. We believe that our work opens many research opportunities for the future: designing switch architectures that implement learned routing protocols, using `Mistill` as the basis for Reinforcement Learning based protocol design, and using learned patterns to optimize existing TE policies, e.g., to reduce the number of exchanged update messages.

## REPRODUCIBILITY STATEMENT

We make all source code as well as the trained models available for the reviewers. Upon acceptance, we will make the code and the models publicly available. To improve reproducibility, we include scripts and manuals on how to recreate figures from the paper.

We will post the link to the anonymous git repository using the discussion forum. Once the discussion forum opens, we will post a link to the repository there with restricted visibility to the reviewers and area chairs.

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

## A HYPER PARAMETER OPTIMIZATION

Finding the parameters was a two stage process. Step one was finding parameters that are able to reproduce the shortest path without any network state info. In the second step, we introduced the network state information.

### A.1 SHORTEST PATH

To learn the shortest paths, we used only the LOCMOD. We experimented with different node representations (see Appendix D), as well as different NN architectures. We experimented with multi-head attention with the IP of the destination as query and the IPs of the neighbors as keys and values. We also tried simple feed forward neural networks. It turned out, that a simple feed forward neural network with one hidden layer and the concatenation of destination IP and the IP of the current location is enough to reliably learn the forwarding behavior in Fat-Tree topologies. We performed a parameter analysis of the hyper parameters with ASHA (Li et al. (2018)) implemented in Ray Tune Liaw et al. (2018). Hyper parameters were:

- batch size: $\{64, \ldots, 256\}$
- hidden layer sizes: $\{8, \ldots, 100\}^3$, i.e., up to three layers.
- learning rate: $[0.001, 0.0001]$.
- samples for ASHA: 100.

We used the ADAM (Kingma & Ba (2015)) optimizer and ReLU activation. The best performing model has one hidden layer with 90 neurons. We thus used one hidden layer with 90 neurons for the LOCMOD and kept them fixed when experimenting with network state.

### A.2 NETWORK STATE

To learn the forwarding behavior with network state, we performed extensive parameter sweeps with ASHA. We explored the following parameter space:

- batch size: $\{64, \ldots, 128\}$
- final FCNs in the FWD MOD: $\{50, \ldots, 128\}^{2,3}$ two or three layers.
- FCN for encoding links: $\{32, \ldots, 150\}$
- Number of blocks: $\{1, \ldots, 65\}$
- Arity: $\{2, \ldots, 16\}$
- Tempererature of $\mathrm{gumbelSoftmax}$: 0.6
- number of heads for MHA in NSMOD: $\{9, \ldots, 14\}$
- dimension of FCN for MHA module in NSMOD: $\{70, \ldots, 129\}$
- hidden dimension of MHA heads: $\{20, \ldots, 64\}$
- output dimension of MHA heads: $\{16, \ldots, 4\}$
- learning rate: $[10^{-4.3}, 10^{-3.5}]$

To obtain the final models, we fixed the arity of the categorical distributions in the HNSAMOD to 2 and the number of blocks to 64. We then trained 100 models with ASHA, resulting in the parameters in Tbl. 1. To evaluate the impact of the number of blocks, we kept the resulting parameters in Tbl. 1 fixed and varied the number of blocks. One area of future work can be a more thorough search of the parameter space to reduce the model size. We did not try to reduce the number of parameters as much as possible. For example, we believe that multiple attention heads can be pruned, as well as the number of blocks be reduced. Those elements of the network showed a large degree of redundancy.

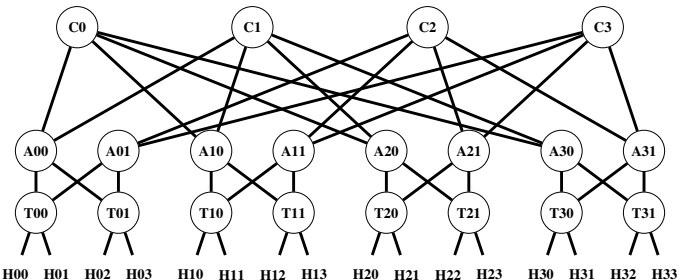

Figure 7: Fat Tree topology - A scalable three-tier Clos network. The topology is a multi-rooted tree with high path redundancy between any pair of hosts.

## B  CLOS-TOPOLOGIES

Data center networks are a good fit to start with learning network protocols. They have regular and repetitive structures and their addressing schemes can encode the location of network nodes. Clos networks are one such common data center topology. One special instance of Clos networks is the (Fat Tree Al-Fares et al. (2008)); a well-known example that employs a three-tier structure and can interconnect are large number of servers using simple commodity hardware. Fig. 7 shows a $k = 4$ Fat Tree topology. A Fat Tree topology has three layers of switches: Top-of-the-Rack, aggregation and core switches. Hosts are connected to Top-of-Rack (ToR) switches, which in turn are connected to aggregation switches. ToR and aggregation switches are organized in pods. The Fat Tree in Fig. 7 has four pods which are interconnected by four core switches. A Fat Tree can support $\frac{k^3}{4}$ server nodes with $\frac{5k^2}{4}$ $k$-port switches. For instance, a Fat Tree constructed from 8-port switches can support 128 hosts with 80 switches. Clos networks provide a rich, high-capacity connectivity matrix, i.e., $\frac{k^2}{4}$ paths exist between any pair of servers in different pods, and $\frac{k}{2}$ paths between servers in the same pod that are connected to different ToRs. The Fat Tree in Fig. 7 already provides 4 paths between servers in different pods, and 2 paths between servers in the same pod connected to different ToRs.

Clos networks have two advantages: Simplified inventory Management and resilience to failures (Al-Fares et al. (2008); Dutt (2017)). Inventory management is simplified because the network can be build from fixed-form switches. This makes it easy to stock spare devices, and replace failed ones, since each switch is the same. Resilience of the topology is increased through high path diversity. If a link or device fails, enough alternative routes exist.

## C  DEPLOYING DISTILLED PROTOCOLS

This section outlines two potential practical implementations of `Mistill`. Sec. C.1 outlines an implementation of `Mistill` on switches in the network. Sec.C.2 outlines an implementation of `Mistill` in the edge of the network. Finally, Sec. C.3 describes how update messages can be send and details how the resulting overhead can be computed.

### C.1  IN-NETWORK DEPLOYMENT

In this deployment, `Mistill` is implemented in the network without touching the servers. Also, `Mistill` does not change the data plane traffic. Thus, `Mistill` is transparent to the server nodes, and can be deployed incrementally in the network.

To be operable, `Mistill` needs routines and data structures on switches that interface with the NN. Each switch has four entities: A Flow(let) Table (FT), the Network State Database (NSDB), the NN, and a Receiver Database (RDB). Fig. 8 shows the entities and their interaction.

The NSDB stores `HNSAs` received from other switches in the network. The switch populates the NSDB with the content of `HNSAs`. Switches do only need `HNSAs` from parts of the network flows

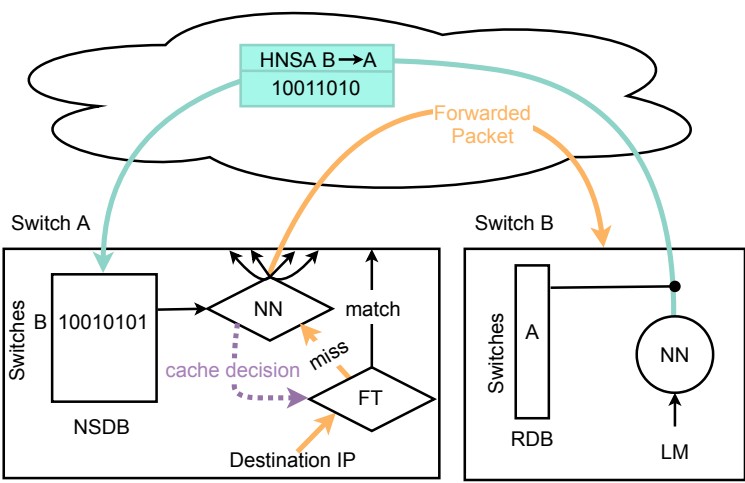

Figure 8: Realization of `Mistill` on a switch.

traversing the switch are traveling to. `HNSAs` from other parts of the network are irrelevant, as Fig. 6a illustrates.

The NN is used in two functions on the switch: 1) to compute forwarding decisions, and 2) to compute the `HNSA` content. To compute forwarding decisions, the switch needs information from the packet. In our examples, this amounts to the source and destination IP address. All other inputs reside in the memory of the switch. To compute the `HNSA` content, the switch monitors the required statistics, e.g., the link utilization or queue sizes. The switch executes the HNSAMOD with the current link state. The resulting `HNSA` is propagated through the network to other parts of the network as indicated by the RDB. The propagation of `HNSAs` is detailed in Sec. C.3.

The FT caches forwarding decisions to avoid packet-reordering at the destination. We believe that it is likely that the content of the NSDB is updated frequently during the life-time of longer flows. As a result, it is likely that packets of the same flow are forwarded along different paths in the network. Thus, the switch caches the forward decision the NN produces for the first packet of a flow(let) and re-uses this decision for subsequent packets. This approach additionally decreases the computational effort. Caching of forwarding decisions to avoid packet re-ordering is also performed by other in-dataplane TE solutions (Alizadeh et al. (2014); Katta et al. (2016); Hsu et al. (2020)).

The RDB controls to which nodes in the network `HNSAs` are send. The RDB is lazily populated based on the source IP addresses of packets the switch observes during operation. Sec. C.3 explains the sending of `HNSAs` in detail.

To guarantee delivery of packets, the FT can be used to configure static routes, e.g., to implement ECMP. If there is a training artifact in the NN that leads to wrong forwarding decisions, then affected traffic can use the fallback routes once the Time to Live (TTL) reaches a certain value. In our experiments, we did not experience any loops, though.

In this way, `Mistill` can be deployed incrementally. For example, `Mistill` can be deployed first on the ToR and aggregation switch of individual pods. Depending on the results, the number of pods that are operated with `Mistill` can be gradually increased. This approach requires the configuration of static rules to let legacy pods that do not run `Mistill` interact with pods that run `Mistill`. This is necessary, since `Mistill` switches will not use the existing routing protocol. Thus, the legacy switches need default routes to those pods. Switches that run with `Mistill` do not need static routes, they can be learned during training.

## C.2 EDGE-BASED DEPLOYMENT

The in-network based deployment requires that switches have substantial computational requirements. For example, switches that are based on FPGAs (Luinaud et al. (2020)), or have hardware

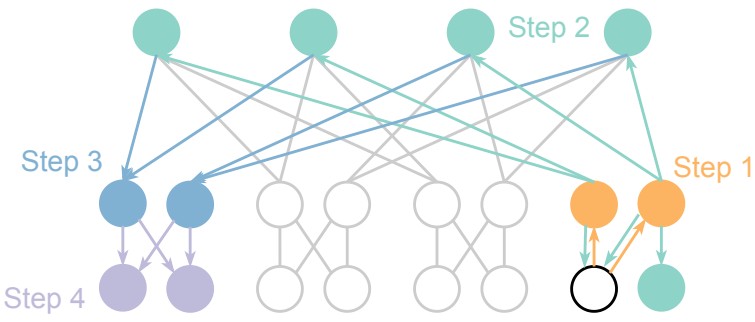

Figure 9: Probing scheme for all node types.

support for executing ML models (Swamy et al. (2020)). Such switches are not available in existing data-centers. Thus, `Mistill` cannot be deployed there. Instead, we suggest a hybrid implementation, where only the `HNSAs` are calculated on the switches, and the forwarding logic is moved into the servers.

Our results show that the computation of the `HNSAs` does not require a complex NN. The forward pass of this network takes less than one millisecond on an Intel(R) Core(TM) i7-7700HQ CPU with 2.80 GHz. Thus, the NN could operate in the control plane of the switch. The switch can calculate the `HNSA` content there and send packets to the end-hosts to inform them about its state. Running this process in the control plane of the switch might impact the frequency with which updates can be send, since the communication between control- and data-plane takes time. Evaluating the potential performance is one area of future work.

The computation of the forwarding decisions is moved into the end-hosts. For each new flow, the end-host then determines a path through the network. The end-host calculates the aggregation switch its ToR switch should send the packet to. Then, the host can calculate the core switch the aggregation switch should send the packet to. From a core switch, only one path exists to the destination host. Forwarding of the packets can be realized, e.g., with the Multiprotocol Label Switching (MPLS) protocol. The end-hosts add MPLS labels to the packets that determine the path through the network. Switches forward the packets based on those labels. MPLS is a broadly available feature in data center equipment.

The required functionality on the end-hosts can be implemented with the extended Berkeley Packet Filter (eBPF) Calavera & Fontana (2019). eBPF is a standard Linux technology and also available in Windows starting with Windows 10 and Windows Server 2016. eBPF allows the execution of custom byte-code in the Linux kernel. The execution of eBPF bytecode is triggered by certain hooks in the kernel. In this way, the forwarding logic of `Mistill` can be invoked for outgoing packets. For a packet of a new flow(let), a new route is calculated and stored in the NSDB. If an entry exists, the calculated route is used instead. Similarly, `HNSAs` from the switches can be processed in the kernel with eBPF. Further, SmartNICs that can accelerate the execution of NNs exist for end-hosts today (Sanvito et al. (2018)). Thus, the calculation of the NN and the processing of `HNSAs` can be offloaded to the NIC (Vieira et al. (2020)).

### C.3 HNSA PROPAGATION

We design an efficient propagation mechanism based on multi-casting that is inspired from the learned pattern in the SI and takes advantage of the structure of Clos-topologies. The propagation mechanism keeps the number of `HNSAs` that circulate in the network in the order of $\mathcal{O}(N)$, where $N$ is the number of switches in the network.

The propagation mechanism exploits two observations: 1) up-stream switches need update messages from down-stream switches to make forwarding decisions. 2) Switches in one pod need update messages from a similar set of down-stream switches, since each switch is likely to see a similar set of destination IPs. Thus, we propose to multi-cast a `HNSA` to all switches in the source pod. `HNSAs` have a header with two fields 1) the IP of the switch from which the `HNSA` originates and 2) a bit

array of length $k$, where $k$ is the number of pods. Each switch tracks the pods from which traffic passing them originates, i.e., the RDB. The source pod of a packet can be deduced from the source IP-address (Al-Fares et al. (2008)). The switch indicates those pods in a bit array stored in the `HNSA` header.

Fig. 9 shows an exemplary `HNSA` propagation from a ToR. First, the ToR sends its `HNSAs` to all neighboring aggregation switches (Step 1). The aggregation switches forward the probes to all ToR switches in the pod as well as all adjacent cores switches (Step 2). The ToR switches consume the probes and do not circulate them further. The core switches forward the `HNSA` to all adjacent aggregation switches in pods indicated by the bit array in the `HNSA` header (Step 3). The aggregation switches then forward the `HNSA` to *only* the incident ToR switches, i.e., do not send it back up into the core layer (Step 4). For this, the aggregation switches check for each `HNSA` that they receive whether the `HNSA` originated in their own pod. If it originated in a different pod, then the aggregation switch does not forward the probe to the core layer. The ToR switches do not further circulate the `HNSA`. The checks if a `HNSA` originated in the same pod or not can be performed by comparing the IP address of the source switch with the IP address of the switch that is currently processing the `HNSA`.

Further, switches forward probes only, if they did not receive a probe from the switch identified by the source IP within a configurable amount of time. This can be achieved at line rate (Katta et al. (2016)). Thus, aggregation switches in the source pod will forward only one probe to their connected ToR switches.

The overhead of this probing scheme is reasonable. In the worst case, all pods communicate with each other. at most $\frac{5k^2}{4}$ messages will traverse one link. One message for every switch in the network. Assuming a probing frequency of $1\,\text{ms}$ and a link bandwidth of $10\,\text{Gbits}$, the probing overhead is then:

$$\frac{\frac{5k^2}{4}(54\,\text{Byte} \cdot 8 + k\,\text{Bit})}{1e^{10}\,\text{Bits} \cdot 0.001\,\text{s}}. \tag{1}$$

We assume a size of $46\,\text{Bytes}$ plus $k\,\text{Bit}$ for the update message, consisting of $14\,\text{Bytes}$ Ethernet header, $20\,\text{Bytes}$ IP header and $12\,\text{Bytes}$ plus $k\,\text{Bit}$ for the probe. The $12\,\text{Bytes}$ of the probe consist of $4\,\text{Byte}$ for the source IP, $8\,\text{Byte}$ for the `HNSA` content, and $k\,\text{Bit}$ to indicate the pods the `HNSA` should be forwarded to.

Thus, for a $k = 8$ Fat-Tree, the overhead of probing one one link would be $0.3\,\%$ of the available bandwidth. For a $k = 16$ Fat-Tree supporting more than $1\,000$ hosts, the overhead would still be $1.22\,\%$.

However, the evaluation of the SI in Sec. 4.4 shows that e.g., the `HNSAs` of core switches are not requested by any other switch. Further, if not all pods communicate with each other, as e.g., the case in the `render` or `combine` pattern, then the number of `HNSAs` will further decrease. In practice we expect the number of update message thus to be less.

## D    NODE ADDRESSING SCHEMES

Addresses in communication networks should follow a pattern to, e.g., keep routing tables small (Waldvogel et al. (1997)). If IP addresses did not contain a useful pattern, routing tables would have to store forwarding decisions for each and every IP address. Since the addresses are written to the header of every packet, they should also be concise to reduce overhead.

IP addresses today are geared towards the available technology, e.g., towards longest prefix matching. It is unclear if the way nodes are addressed today is a good fit for routing protocols learned with ML.

Thus in this section, we investigate whether ML based routing protocols can be learned with IP addresses, whether latent space models, as used in natural language processing or graph learning, yield an addressing scheme that results in better performance, and whether the addressing scheme plays a role at all for the ability to learn a routing scheme.

### D.1 LEARNING BINARY EMBEDDINGS OF NODES IN GRAPHS

Learning useful representations for nodes in graphs is subject to ongoing research (Hamilton et al. (2017)). We use Bernoulli Embeddings (Misra & Bhatia (2018)) to learn node addresses. Bernoulli Embeddings map nodes in a graph into a binary space that encodes the neighborhood relationship. We do not consider approaches such as `node2vec` (Grover & Leskovec (2016)) that embed nodes into a continuous latent space as the size the necessary floating point numbers is prohibitive. We do also not consider methods that use additional meta-data for generating embeddings. Meta-data on the networking level, e.g., MAC addresses, hardware information, etc. is either random or not correlated with the network structure.

A binary embedding is a $d$-dimensional binary vector, i.e., the embedding of node $i$ is $E_i \in \{0,1\}^d$. The binary vector is sampled from $d$ independent Bernoulli distributions, parameterized by a matrix $\Theta \in \mathbb{R}^{|\mathcal{V}| \times d}$. The $k^{\text{th}}$ element in the embedding of node $i$ is thus sampled according to: $E_i^k \sim \text{Bernoulli}(\Theta_{i,k})$. The embedding can be optimized by changing the parameterization of the independent Bernoulli distributions such that an appropriate loss function is minimized.

The intuition behind the method in (Misra & Bhatia (2018)) is to encode neighborhood relation into the embeddings by casting the problem as a link prediction problem: The probability of an edge $e_{ij}$ from node $i$ to node $j$ is proportional to the hamming distance $d_H$ between the embeddings $E_i$ and $E_j$ of those nodes:

$$p(e_{ij} \mid E_i, E_j) = \frac{\exp\left(-\alpha d_H(E_i, E_j)\right)}{\sum_{k \in \mathcal{V}} \exp\left(-\alpha d_H(E_i, E_k)\right)} \tag{2}$$

The parameter $\alpha$ is a scaling factor. The hamming distance $d_H$ between two vectors is defined as: $d_H : \{0,1\}^d \times \{0,1\}^d \to N; x, y \to x^T(1-y) + (1-x)^T y$.

The embeddings can then be trained by optimizing the log probability, i.e.:

$$\mathcal{L}(G, \Theta) = - \sum_{(i,j) \in \mathcal{E}} \mathbb{E}\left[\log p(e_{ij} \mid E_i, E_j)\right]_\Theta, \tag{3}$$

assuming that edges are independent of each other given the embeddings of incident nodes.

Misra et al. (Misra & Bhatia (2018)) use a continuous approximation to the objective in Eq. (3) to circumvent the expectation over a discrete valued variable. In contrast, we make use of the Gumbel-Softmax trick (Jang et al. (2017); Maddison et al. (2017)), a reparameterization trick for discrete random variables, to optimize the objective in Eq. (3). In addition, Misra et al. (Misra & Bhatia (2018)) use Noise Contrastive Estimation (NCE) to avoid the summation over all nodes in the graph in the denominator of Eq. (2). We do not use NCE since optimization is feasible for the size of communication networks. In addition, we experienced that NCE makes learning of unique embeddings more difficult due to aliasing effects between nodes.

Addresses in communication networks must be unique which translates to the embeddings of the nodes. The optimization objective in Eq. (3) is not designed for this constraint and might assign the same embedding to different nodes. In fact, for a fully connected graph a single embedding for all nodes would in fact be the optimal solution. Also in other networks nodes exists that are structurally equivalent and thus get the same embedding. For example all hosts connected to one ToR switch in a Fat-Tree topology would get the same embedding. Similarly, all core switches in a fat-tree that connect to the same aggregation switches are structurally equivalent and would get the same embedding. We thus adapt the objective in order to avoid duplicate embeddings. Specifically, we use the following energy function for the model of edge probabilities:

$$\mathcal{L}(G, \Theta) = -\alpha \sqrt{\left(d_H(E_i, E_j) - \beta\right)^2}, \tag{4}$$

where $\beta$ is a scalar parameter. The intuition behind Eq. (4) is that a probability of one for an edge is obtained only, if the embeddings are different. The parameter $\beta$ controls the scaling of the difference. For example, $\beta = 1$ pushes the embeddings of adjacent nodes towards having a hamming distance of one.

### D.2 RESULTS

The suitability of addressing schemes is evaluated by learning static shortest paths. An addressing scheme is suitable if the forwarding rules for static ECMP can be learned reliably, i.e., if we obtain

| Optimal | Random | Learned12 | Learned16 | Learned20 | Learned24 | Composed | IP |
|---------|--------|-----------|-----------|-----------|-----------|----------|--------|
| 0.3707  | 0.6210 | 0.5397    | 0.4835    | 0.5002    | 0.6541    | 0.4185   | 0.3921 |

Table 2: Training results on the shortest path training tasks for graphs and embeddings. For the learned embeddings, different vector sizes were evaluated. Optimal is the average loss value for the ground truth.

a low loss value. Tbl. 2 lists the results obtained for this learning task. We use addresses of up to 24 bits. IP addresses follow the scheme in (Al-Fares et al. (2008)). We remove the first octet since it is constant and thus does not contribute any information.

The random embedding results in the highest, i.e., worst, loss value. The addresses do not contain any form of pattern, after all. The learned embeddings result in lower loss values. The dimensionality of the embedding does not have an effect on performance. No consistent increasing or decreasing trend is observable. The IP address based embedding results in the lowest loss values. Good performance here is explainable, since the forwarding behavior can be perfectly computed from the IP address alone in case of the Fat Tree. The composed embedding for the Fat Tree achieves a loss value of $0.4185$ and is competitive with the IP-based result of $0.3921$. This hints at filtering out similar structures in the graph and assigning embeddings to those structures in an equal manner.

In summary, IP addresses are well suited to learn forwarding decisions on well structured topologies such as Fat Trees. This has the advantage that learned routing protocols are backwards compatible, can be incrementally deployed, or run side-by-side with traditional routing protocols. For `Mistill`, we thus use the last $24$ bits of the IP addresses to represent nodes.

