# OpenReview forum: "Mistill: Distilling Distributed Network Protocols from Examples"
_ICLR.cc/2022/Conference — ICLR 2022 Submitted_

### Official Review · Reviewer_wBxW · 2021-11-01

**Correctness:** 4
**Technical Novelty And Significance:** 3
**Empirical Novelty And Significance:** Not applicable
**Recommendation:** 6
**Confidence:** 4

**Main Review:**

The paper studies the use of ML to design protocols for a data-center application. In particular, a single-server data center, a Fat-Tree topology, and a single policy are assumed. A neural network is training to learn forwarding policies from examples.  A modular structure is proposed to minimize complexity. The NN is trained to be adaptive, but the model itself is fixed at the end of training.

The paper is a welcome addition to the sparse set of papers on applying ML to problems in communication networks (compared with image classification, for example). A challenge, of course, is availability of data.

Overall, the paper reads well, and it contains detailed information on the architecture and training (perhaps a bit too much for the ICLR audience).

There are several issues that need clarification:

(1) Do all switches need to have MISTILL installed? Can this work in a hybrid architecture where some switches do not have MISTILL?

(2) The work is focused on data-center application; the assumption of Clos-topologies makes it inapplicable to packet-switched networks, which are the norm today.

(3) Is the topology assumed known? Does every node know its parents and children nodes?

(4) The specific policy learned here is that of dropping packets or flows if a node is unreachable due to link or node failures; this seems to be a simple problem given the fat tree structure. Does the policy also switch to a new route if the current route is unavailable?

(5) Typically protocols must instantiate several policies, not just a single policy. Can the framework be extended to do that?

(6) Given that the topology is assumed to be a tree, it is not clear that the approach can be extended to making use of new links, as would be the case in an ad hoc mesh network (not just links and hence paths disappearing in a tree, but also new links forming or potentially forming).

(7) A \beta term is introduced in the modified loss function in (6) to ensure that embeddings corresponding to different nodes do not end up close to each other. But this is precisely what contrastive learning is supposed to achieve. It is not clear why the authors were not able to get NCE to work.

(8) Associated with nodes and links is context information which can also be used to differentiate between embeddings. This has been widely used in the social networks literature. This paper assumes that nodes are identical and have no context information.

 (9) The paper makes a criticism of CONTRA that the latter relies on the use of a high-level policy language.  What is wrong with that approach? If a ML model is to learn a policy, it should be able to recognize and resolve conflicts in the examples provided (the conflict may be due to lack of full context being provided or used by the learner). A purely data-driven approach may not be able to offer insights on what has been learned, vs. a neuro-symbolic approach where rules might be learned, and possibly interpretable by a human.

(10) Figures 5 and 6 provide some hints. But it is not clear what the model has learned.

(11) There is some discussion of overhead (top of page 5), but this is insufficient. Would like to get some more details.

(12) Shortest path routes are mentioned several times; but there is only one cycle-free path on a tree.

(13) The 2009 Al-Fares paper indicates that a k-fat tree has (k/2)^2 k-core switches and can support k^3/4 hosts. For k=8 this leads to 16 switches and 128 hosts, but the paper says 80 switches and 128 hosts. Please clarify. Also are there k=8 servers at the data center?


Typos and such
- Please define acronyms before, not after, first use; e.g. HNSA in Sec 3



**Summary Of The Paper:**

The paper studies the use of ML to design protocols for a data-center application. In particular, a single-server data center, a Fat-Tree topology, and a single policy are assumed. A neural network is training to learn forwarding policies from examples.  A modular structure is proposed to minimize complexity. The NN is trained to be adaptive, but the model itself is fixed at the end of training.

**Summary Of The Review:**

The paper studies the use of ML to design protocols for a data-center application; it adds to the sparse set of papers on the application of  ML to problems in communication networks, and is a welcome change from papers dealing yet again with MNIST.

My main concern relates to generalizability: Can the framework be extended to non-tree topologies (in particular, where links may appear as well as disappear, and where multiple routes exist), to protocols that must support multiple policies, and to incorporation of contextual information associated with nodes (perhaps also links). It is not clear what the model has learned, and some additional details about the test setup are needed.

---

> ### Author Response · Authors · 2021-11-15
> **Response to Reviewer wBxW**
>
> Thank you very much for your thorough review. We address your points individually in the following:
>
> 1) A hybrid deployment is partly possible. Existing protocols such as BGP or OSPF cannot be deployed. Existing hardware might be usable if static rules can be injected. We will add this to the discussion.
> 2) The used topology as presented by (Al-Fares et al. 2008) has a Clos-structure but uses packet switches. It has been specifically designed with the datacenter use-case in mind.  We will include a section in the background that presents the considered topology more in detail to give a better intuition on the properties of these topologies and how they are operated. Clos-topologies are a specific type and an extension to more general networks is an important next step.
> 3) The topology is assumed to be known at training time and to stay fixed during the duration of deployment. Once deployed, the switches do not need to know the topology of the network. At most, they need to know their direct neighbors to correctly associate the output ports with the outputs of the NN. We will add a remark.
> 4) The NN learns to drop packets if no route to a destination exists. If routes exist, the NN takes the one that is best under the policy, e.g., the path with the smallest maximum link utilization. In case a link fails, the NN can also reroute around this failure. For example, if the link from an Aggregation switch to a ToR switch breaks, then the NN can reroute packets via another ToR and another Aggregation switch to the destination. Dropping packets if no route to the destination exists looks simple but is hard in practice. Existing TE algorithms (including CONGA) are not able to do this, since they miss the last hop. They would always forward traffic to the ToR the destination should be connected to, where it is then dropped. We will add a corresponding remark.
> 5) It might be. A solution could be to train multiple heads and then select the ones that should be used now. We leave this open for future work as this paper puts the focus on whether we can learn distributed protocols at the first glance.
> 6) This is correct. We assume a static topology that does not change except for link failures, which is the case for current datacenter networks. An extension to general networks is an important next step.
> 7) NCE did work, the embeddings were not useful for the downstream task, though. We included the additional term to push structurally equivalent nodes away from each other. Without this, nodes frequently got the same binary embedding, since they were structurally equivalent. Of course, that is what the objective should do to begin with, so its perfectly fine. This behavior kills networking though, since you have multiple nodes with the same destination address. To which one should you send the packet?
> We will explain this intuition in the appendix.
> 8) This is correct. To replace the context information, we added the beta term in the objective. Also, the context information is difficult to obtain in Mistill, since here all context information would be discrete and different for each node, like MAC addresses, or identical, or not correlated with the connectivity information and, thus, would not be useful for TE. Due to space restrictions, we will discuss this in the appendix.
> 9) CONTRA's approach is fine. In contrast to Mistill, it is limited by the policy language. For this, you get a lot of benefits. But what happens if you want to express something that does not work with the policy language? For example, you know the size of a flow and based on that and the current utilization on this node and the neighbor you decide where to go? That is where Mistill comes in. The problem with the rules is that there might be quite a lot and then you run into the table space problem. The idea of Mistill is to use a connectionist approach and to store the patterns in the NN instead of in explicit rules.
> 10) Thank you for this suggestion. Unfortunately, it is a bit vague to us. Could you elaborate a bit more? Do you refer to an interpretation of, e.g., the attention scores, i.e., switch A needs state from switch B to make decisions for Host C?
> 11) Thank you for this suggestion. We will add more details on involved overhead of the probing in the appendix.
> 12) We will address this issue through a background section explaining Clos-topologies to show that they are multi-rooted trees and there are k^2/4 shortest paths between any pair of hosts in different pods.
> 13) Thank you for this question. As you point out, k=8 leads to 16 core switches. These are connected via an aggregation and edge (ToR) layer to the hosts. Each of these layers consists of 32 switches. This gives in total 80 switches in the topology. The hosts are connected to only one ToR switch. Thus, they need only one port. As for points 2) and 12), we will extend the description of the topology.
>
> We will also carefully check the manuscript for defining acronyms before first use.

---

> > ### Comment · Reviewer_wBxW · 2021-12-02
> > **Comment on Authors' responses**
> >
> > The authors have satisfactorily addressed all my comments.
> > My (10): From the paper it is not clear what the model has learned - can we interpret what has been learned? I have revised my scores

---

> > > ### Author Response · Authors · 2021-12-02
> > > **Comment**
> > >
> > > Thank you for your response and clarification of your question. If the paper gets accepted, we will emphasize this point more in the camera-ready version.
> > >
> > > Indeed, it is possible to interpret what the NN has learned: The NN correctly learned whose HNSA's are needed to cover the edges between the switch making a decision and the destination. The attention weights are reminiscent of a full edge-cover. Covering all edges on paths between a switch and the destination node is reasonable in our setting since we use random link weights and random link and node failures.
> > >
> > > Figure 6a illustrates this behavior. Figure 6a shows the attention weights for all switches for host H-10 as the destination, i.e., the first host connected to the first Top of Rack (ToR) switch in the second pod (see Figure 7 in Appendix B). The first row of Figure 6a contains the attention weights of ToR-00, i.e., the first ToR switch in the first pod. To make a forwarding decision for H-10, ToR-00 needs HNSAs from itself, ToR-01, ToR-11, AGG-00, AGG-03, AGG-10, AGG-11, AGG-12, and AGG-13. The HNSAs from those switches cover almost all edges that form the primary path and the backup paths connecting ToR-00 to H-10. The attention weights closely follow the perfect edge-cover that we illustrated in Fig. 6c.
> > > During our experiments, we also trained NNs where we used the edge-cover in Figure 6c as input, i.e., did not learn the attention weights. The resulting NN could also find the correct paths but was worse than the gumbelSoftmax implementation.

---

### Official Review · Reviewer_hRA9 · 2021-11-02

**Correctness:** 3
**Technical Novelty And Significance:** 2
**Empirical Novelty And Significance:** 2
**Recommendation:** 5
**Confidence:** 3

**Main Review:**

This work tries to apply existing machine learning techniques to solve an important computer network problem; however, I have a number of concerns with this work:

1. The first concern is related to the contribution of the paper. This paper is mainly applying existing neural network techniques for the traffic engineering problem. There is a very limited contribution towards the machine learning techniques. On the other hand, if we focus on the application side, it seems the compute network evaluation environment is not realistic enough since only simulation results are provided (more on this point later).

2. The authors have emphasized in the paper that the network switches have limited computing power and the forwarding model needs to run in the data plane, however, there is no evaluation to show that the designed model would be able to run sufficiently fast in such an environment. In fact, (Chen et al. 2018) have shown that it is challenging to directly use neural networks in the data plane to make forwarding decisions due to latency constraints. Neural network latencies are at a millisecond level, but In today’s data center (which is the main concern for this paper), switch need to make switching decisions at a microsecond level. It is important to evaluate this in a real environment to persuade the reader that this method is actually feasible.

3. One of the challenges when designing TE techniques is the low latency monitoring of the network state (such as in Alizadeh et al. 2018). In this work, the authors propose sharing each switch’s local state in a broadcast manner (selective nodes broadcast), is this an efficient communication pattern, would it cause high latency and overhead.

4. One of the benefits of decentralized control is the scalability of the system. However, it seems the current design requires retraining the model when adding more switches to the network.

5. What is the traffic load used in the evaluation, would the algorithms still work when different traffic load is used to evaluate the system? The policies used in the evaluation are relatively simple, more realistic workloads would be more desirable.

6. Presentation issue: some of the images’ legends are overlapping with the curve.


**Summary Of The Paper:**

This work is targeting the problem of traffic engineering in computer networks. It proposes a learning method called Mistill for making packet forwarding decisions in distributed network switches using data generated by a centralized policy. The paper assumes that a forwarding policy can be obtained in a centralized fashion with global information, and the goal is to train a neural network model that can make the correct forwarding decision in each switch without global states. The network switch can communicate its local state to other switches to improve the overall performance.


**Summary Of The Review:**

This paper applies existing machine learning techniques to solve the traffic engineering problem in computer networks. The contribution toward the machine learning methodologies seems to be limited in this work, and the evaluation needs to be conducted in a more realistic environment to demonstrate the feasibility of this proposed method for the targeted application.

---

> ### Author Response · Authors · 2021-11-14
> **Response to Reviewer hRA9**
>
> Thank you very much for your thorough review. We address your points individually in the following:
>
> 1. Indeed, our first evaluations rely on simulations only, which abstract switch processing times. However, our focus point is to answer the question whether a neural network-based approach can learn routing policies such as ECMP in a datacenter environment at all. Hence, we focus on simulations because evaluating addressing schemes and forwarding decisions themselves will not be impacted by hardware limitations such as computing power. Of course other factors such as CPU or forwarding pipelines of switches can impact the overall forwarding latencies. However, taking such effects into account is a future next step.
>
> 2. Thank you for raising this concern. Indeed, full NNs are not yet deployed to switches. However, our concept is also not limited to in-network processing only. For instance, Mistill might be deployed in an edge-based manner. We will discuss this in an Appendix about the realization. In such case, Mistill can still benefit from fast computing hardware.  The idea revolves around making a decision not per packet but per flow which has additional advantages. This reduces the computational overhead.
>
> 3. This is an aspect of every distributed routing protocol to exchange local state. Of course, this can be problematic if excessive amounts of data are exchanged. Broadcasting the state is a common approach, e.g., OSPF (https://datatracker.ietf.org/doc/html/rfc2328) or HULA (Katta et al. 2016). Mistill learns which state information has to be exchanged. In the revised version, we will discuss more clearly that broadcasting link state information is a well established approach.
>
> 4. Your observation is correct. Increasing the network size requires to retrain the network. However, as a recent publication shows (https://arxiv.org/abs/2110.08374), traffic changes that require topological changes of the network happen on a coarse time-scale (~weeks/months). Retraining Mistill’s model takes approximately one day, hence, is feasible on the required scale. In addition, extensions of the network are planned operations such that re-training can be conducted beforehand. When the new switches become active, the updated model only needs to be rolled-out; recall, that Mistill’s training data can be generated using simulations.
> We will add a discussion of this aspect in the revised manuscript.
>
> 5. Thank you for this suggestion. The present evaluation uses link utilization values that are sampled uniformly at random.
> We are currently running three more scenarios based on traffic distributions from Facebook’s network (https://dl.acm.org/doi/10.1145/2829988.2787472). The initial results confirm the conclusions from the evaluation on the uniformly randomly generated load.
> The evaluation is added to Section 4 in the revised manuscript.
> ECMP and WCMP (https://dl.acm.org/doi/abs/10.1145/2592798.2592803) are well established routing policies both in production as well as a baseline in research. Therefore, they are an interesting target for Mistill.
> Similarly, LCP and MinMax also established in related work, e.g., CONGA (Alizadeh et al. 2014). These policies route optimally given the load in the network. However, they require global knowledge of the current link weights and are harder to implement than ECMP.
>
> 6) Thank you for this comment. In the revised manuscript, we will improve the figures’ quality.

---

### Official Review · Reviewer_noo6 · 2021-11-03

**Correctness:** 3
**Technical Novelty And Significance:** 3
**Empirical Novelty And Significance:** 3
**Recommendation:** 5
**Confidence:** 4

**Details Of Ethics Concerns:**

No ethical concern.

**Main Review:**

Strengths.
1. The paper gives some insights in incorporating machine learning techniques into the Traffic Engineering area, including how to encode the link-state using reparameterization and how to communicate using attention techniques.

2. The paper gives a general analysis of the pros and cons of ML approaches. It is informative when evaluating the feasibility of applying ML on real switches.

Weaknesses.
1. The paper represents the challenges in a vague way. In Section 1 Introduction, the paper illustrates the necessity of customizing Traffic Engineering schemes for various applications. However, it lacks concrete analysis of challenges that lie in translating a TE policy into a distributed protocol. Only a general description in Paragraph 2, such as "it requires the specification of exchange data, the processing of the data and the algorithm ...", is not solid and analytical enough to demonstrate how these challenge the design of a better TE scheme. The paper should show how difficult it is that to process per-application-level data and to design an algorithm. Furthermore, they do not explain how incorporating machine learning techniques mitigate those challenges.

2. The paper presents unclear descriptions of technical details. 1) In the second paragraph of Section 2, they claim that the proposed approach will learn two aspects, i.e., process and exchange the local state of switches and map the exchanged state into forwarding decisions. However, later in the fourth paragraph, they mention that the model should learn more than these two aspects. The model should also learn how to react to changes in the network, such as node and link failures and changes in monitored measures. They do not explain how the latter aspect is related to the former two aspects. 2) In Section 3, paragraph 5, does "to select the HNSAs necessary for this" indicates that there might be more HNSAs? Also, it is unclear what "this" refers to.

3. The evaluation part is not complete enough. First, the paper does not show the comparison of computing overhead compared to baselines. As a result, we do not know if the proposed one can be deployed in actual data center deployment. How long will it take for training? What's the computing overhead for inference? Second, they do not use traffic features from various applications, and thus we do not know if the proposed approach can adapt to various traffic features. It is crucial in Traffic Engineering, as stated in Section 1 Introduction.

4. The paper does not discuss the possible technical defects of their approach. For example, with wrong forwarding decisions accumulating, network congestion may happen. How does the approach handle problems that arise from intrinsic errors of the Machine Learning algorithm?

**Summary Of The Paper:**

The paper proposes an ML-based approach Mistill to help automatically deploy distributed protocol from a given TE policy. The approach learns forwarding decisions together with intermediate results such as exchange information and LinkState encoding from exemplary policies. The resulting network can later be deployed in switches to encode link-state and compute forwarding decisions. The technical contributions are the following. First, they leverage a reparameterization trick to handle categorical distributions when encoding link states into a binary vector. Second, they use the scaled dot-product attention method to help switches learn to exploit the encodings mentioned above and make the forwarding decision. In the evaluation part, the paper compares with three custom baselines, i.e., LCP, MinMax, WCMP, and exceeds all of them in terms of corresponding metrics.

**Summary Of The Review:**

The paper gives good insights into combining machine learning techniques and traffic engineering. However, it lacks solid evaluation to demonstrate the overhead of this approach and lacks analysis of problem handling. Moreover, the representation of background and some of the technical details does not help readers to understand. Therefore, the paper should 1) add evaluations demonstrating the overhead of ML techniques, analyze the possible problems that may arise from prediction errors; 2) rewrite ambiguous sentences to provide more clear technical details; 3) show more concrete difficulties in translating traffic engineering policy into protocols.

---

> ### Author Response · Authors · 2021-11-19
> **Response to Reviewer noo6**
>
> Thank you very much for your thorough review. We address your points individually in the following:
> 1) We agree that the analysis of the challenges is high-level. On the other hand, a quantitative analysis of designing and implementing a distributed protocol itself is challenging. We would be grateful if you could give us a more concrete description of your expectations towards such an analysis.
> For now, we better explain the challenges in the design of the network state processing, information exchange, and forwarding decision calculation. Further, we concede that we did not adequately address how ML helps to overcome these challenges. We will improve this aspect in the next revision.
>
> 2) Thank you for pointing out the unclear formulation. Learning to react to node failures, link failures, and changes in the monitored measures relates to processing and exchanging the local state of switches as follows. A link failure, the failure of a neighboring node, or a change in the monitored measures on switch A can result in different forwarding decisions on a distant switch B. Thus, switch A must inform switch B about its state. To update switch B, switch A must encode its local state into an update message, which we call HNSA in our work. Switch B processes the HNSA of switch A and the HNSAs of other switches to make its forwarding decision.  MISTILL automatically learns the content of the messages, i.e., how to encode the switch local state, which nodes have to exchange update messages, and how to process the update messages of nodes to make a forwarding decision. Each switch creates one HNSA and sends it to switches that need its HNSA. Each switch uses HNSAs of multiple other switches to make forwarding decisions. We revised Sec. 2 for clarity in the revision. Further, we added a detailed description of the processing inside of switches that run MISTILL in the appendix.
>
> 3) We concede to this point. We added training and inference times in the present revision and included four new traffic patterns in the evaluation. We further emphasized the scope of MISTILL, i.e., that MISTILL is not intended as a replacement for existing, highly optimized TE protocols but instead aims at simplifying and enabling the creation of new TE protocols.
>
> 4) We thank the reviewer for pointing out this missing piece of information. Our evaluations did not experience this kind of problem, i.e., all traffic arrived at its destination nodes. However, we agree that a fallback solution might be necessary for practice. A simple solution could be the configuration of static routes that get active once a packet's Time To Live (TTL) falls below a threshold. We will add a remark in the central part of the paper and elaborate in the appendix.

---

### Official Review · Reviewer_qhEy · 2021-11-04

**Correctness:** 4
**Technical Novelty And Significance:** 3
**Empirical Novelty And Significance:** Not applicable
**Recommendation:** 8
**Confidence:** 2

**Main Review:**

This paper proposes NN-driven model for designing novel traffic engineering techniques automatically. This paper is missing details on neural architecture, configuration parameters and experiment details. I had major difficult understanding the write up due to these missing difficulties. Let me elaborate:

1. No details on the model: paper does not provide the details of the NN; how many layers, what is the size of the input, how many neurons per layer. What is the loss function and how is it trained?

2. Experiment setup: Why is 10 simulation sufficient? Is random => uniform random? How many concurrent failures? For a large network, 10 simulations of failure events is not sufficient.

3. Model abstraction and retraining of the network: does the proposed model has to be retrained for a different network size and topology? how long does it take to retrain and what is the initial training time ?

4. Simulation: What traffic is being simulated? What are the datacenter applications that paper is using?

Overall the paper is very light on details and is not fit for publication at this time.

Other issues:
--
1. SmartNICs cannot behave like switches so the NN inference they are planning to do has to be done on P4 switches only. Authors should elaborate on how they plan to deploy their model in practice?
2. Nocomparison with Contra (which is state-of-the-art, they do not show how their TE algorithm behaves against contra for the exact same performance goals)

**Summary Of The Paper:**

This paper proposes NN-driven model for designing novel traffic engineering techniques automatically.

**Summary Of The Review:**

This paper is missing details on neural architecture, configuration parameters and experiment details. I am not sure "what" the novelty of the paper is. It is most probably due to the fact that the paper significantly lacks in details.

---

> ### Author Response · Authors · 2021-11-15
> **Response to Reviewer qhEy**
>
> Thank you very much for your thorough review. We address your points individually in the following:
>
> Main review:
> 1.Thank you for raising this point. We will include the loss functions in Section 3. Further, we will add the number of hidden layers and their sizes in Section 4.
>
> 2. We will elaborate the failure pattern more precisely. We are currently running additional simulations to increase the number of runs up to 30. This allows as calculating reliable confidence intervals, which, however, is already possible with 10 runs. The current load generation considers uniformly generated utilizations. In addition, we add three more load patterns.
>
> 3. Thank you for pointing out this missing information. We will include the training time for a model in Section 3 and also remark on the necessity to retrain the model.
>
> 4. We do not use a specific traffic pattern. Instead, we use uniformly distributed random link weights. With respect to learning, this is much more difficult, since utilizations of individual edges are not correlated. Using a specific pattern bears the risk of overfitting to this particular pattern. Here, we show that Mistill can learn protocols even in the absence of any pattern in the traffic itself, i.e., Mistill does not work simply because the pattern is easy. We evaluate three more load patterns based on traces from Facebook's network (https://dl.acm.org/doi/10.1145/2829988.2787472). We can already observe that Mistill works also for these patterns. We will add this evaluation in the revised manuscript.
>
> Other issues:
> 1. This point is correct. We named SmartNICs to emphasize that there are developments in the equipment of communication networks. SmartNICs allow the acceleration of NNs in the data plane. We believe that this will also be integrated into switches in the future. We will reformulate this remark to emphasize this point. Further, in addressing comments of other reviewers, we include a section in the appendix that discusses potential realizations of Mistill.
>
> 2. We do not compare Mistill with CONTRA or other TE schemes, which, of course, exist. We believe that such evaluation adds little benefit at this point. We do not claim an increased performance of Mistill over CONTRA for any objective. Mistill is not a replacement or will outperform existing highly engineered and optimized TE algorithms. Instead, Mistill uses NNs to learn TE policies. To show that the learned policies have a high quality, we compare the performance of the learned protocols with the optimal solution that uses the global network state. Further, Mistill is one way to simplify the generation of distributed TE protocols.
>
> The detailed contribution of this work is the design of a NN architecture that can learn distributed protocols from examples. We show that it is possible to train such networks and shed light on how those NNs encode information, and what distributed exchange is required to make a forwarding decision. We believe that this is a contribution that is orthogonal to CONTRA. Further, Mistill opens new lines of future work. Examples could be: Reinforcement Learning for TE that uses our architecture as a basis; the design of new TE algorithms, where Mistill learns the protocol that should be deployed or helps in detecting patterns that can be exploited in the final result. Also, Mistill is a first application that motivates the development of switch designs that use NNs in the data plane.
> To address this point, we discuss the similarities and differences of Mistill and CONTRA in more detail in the related work section.

---

> > ### Comment · Reviewer_qhEy · 2021-11-29
> > **Thank you for addressing my comments.**
> >
> > I thank the authors for addressing all my comments and revising the manuscript based on the comments and suggestions. Although the paper outlines the steps for possible practical deployment in appendix, it still does not have a deployment in place, and it is not clear as to what challenges still may exist in realizing such NN-based TE on switches.
> >
> > I will update my score accordingly.

---

### Author Response · Authors · 2021-11-12
**Rebuttal letter**

We thank all reviewers for their time and valuable feedback. We recognized three recurring topics to improve on in the reviews: A more concise description of the scope and objective of our submission, more technical details on potential implementations of MISTILL, and more extensive evaluations.
We want to use the discussion phase to improve our submission accordingly and outline the changes in this message. Naturally, we welcome any suggestions from the reviewers and the public on the planned improvements. In addition, we will address the concerns of each reviewer in detail in the following days.

We will focus the description of the scope and the objective more on the systematic novelty of MISTILL and the potential of enabling innovation in TE in a new line of research. We will better position MISTILL to existing TE approaches and better describe the use-cases in which we believe that MISTLL will be valuable. We will update the Abstract, Introduction, Section 2, and Conclusion to reflect these changes.

To improve the technical details, we will add the Neural Network dimensions, training duration, and timings for the inference to the beginning of Sec. 3. In the appendix, we will inform in detail on the training procedure and the explored hyperparameters. We recognize that this is valuable information that we should have included right away. Further, we will sketch three avenues to realize MISTILL in practice in the appendix. We kindly ask the reviewers to understand that we do not have a complete prototype at this stage. We believe that the realization of MISTILL is a paper on its own.

We concede to the point that the evaluation of different traffic patterns is necessary. We will use the time during the discussion phase to perform an ablation study. We will evaluate known traffic patterns from the literature. We will include traffic patterns from web-search, data mining [1], and Combine and Render applications [2]. Further, we will better explain the evaluation scenario in Section 4 and perform more simulations to improve the credibility of our results.

We thank the reviewers again for their valuable feedback and are looking forward to the coming discussions.

[1] C. Delimitrou, S. Sankar, A. Kansal, and C. Kozyrakis, “ECHO: Recreating network traffic maps for datacenters with tens of thousands of servers,” in IISWC, La Jolla, CA, USA, 2012, pp. 14–24. doi: 10.1109/IISWC.2012.6402896.
[2] A. Roy, H. Zeng, J. Bagga, G. Porter, and A. C. Snoeren, “Inside the Social Network’s (Datacenter) Network,” SIGCOMM Comput. Commun. Rev., vol. 45, no. 4, pp. 123–137, Aug. 2015, doi: 10.1145/2829988.2787472.

---

### Author Response · Authors · 2021-11-19
**First rebuttal revision**

We attached a rebuttal revision that addresses most of the reviewer's feedback. We highlighted changes to the main text in green color. We further extended the appendix to give more details on the training procedure and technical details of a practical realization of MISTILL. We will continue to improve the revision and are looking forward to your feedback.

---

### Author Response · Authors · 2021-11-22
**Second rebuttal revision**

We added a second rebuttal revision in which we worked on the wording and improved the quality of the figures. We also improved the writing in the appendix. Changes are highlighted in green color.

---

### Author Response · Authors · 2021-11-22
**Changed color to black**

Since the end of the discussion period is close, we changed the textcolor of the changes back to black.

---

### Decision · Program_Chairs · 2022-01-20

**Decision:**

Reject

**Comment:**

Most of the reviewers thought this paper has issues where it could be improved.  There was a range of concerns. Most importantly, several reviewers felt the novelty in the paper was unclear as well as the requirement for more details in the experimental evaluations.